# Quantifying controls on rapid and delayed runoff response in double-peak hydrographs using ensemble rainfall-runoff analysis (ERRA)

Huibin Gao<sup>1,2</sup>, Laurent Pfister<sup>3,4</sup>, and James W. Kirchner<sup>1,2,5</sup>

- <sup>1</sup>Department of Environmental Systems Science, ETH Zurich, CH-8092 Zürich, Switzerland
  - <sup>2</sup>Swiss Federal Research Institute WSL, CH-8903 Birmensdorf, Switzerland
  - <sup>3</sup>Environmental Sensing and Modelling Research Unit, Luxembourg Institute of Science and Technology (LIST), L-4422 Belvaux, Luxembourg
  - <sup>4</sup>University of Luxembourg, Faculty of Science, Technology, and Medicine, L-4365 Esch-sur-Alzette, Luxembourg
  - <sup>5</sup>Department of Earth and Planetary Science, University of California, Berkeley, CA, 94720, U.S.A.

Correspondence to: James W. Kirchner (kirchner@env.ethz.ch)

**Abstract.** Double-peak hydrographs are widely observed in diverse hydrological settings, but their implications for our understanding of runoff generation remain unclear. Previous studies of double-peak hydrographs in the extensively instrumented Weierbach catchment have linked the first peak to event water and the second, delayed and broader peak to preevent water. Here we use ensemble rainfall-runoff analysis (ERRA) to quantify how precipitation intensity and antecedent wetness influence groundwater recharge and double-peak runoff generation at the Weierbach catchment (Luxembourg). The spiky first peak can be attributed to a rapid response directly linking precipitation to streamflow via near-surface flowpaths. Relative to this first peak, the second peak is delayed (peaking ~1.5 days after rain falls), lower (~1/3 the height of the first peak), and broader (declining to nearly zero in ~10 days), and can be attributed to a groundwater-mediated pathway that links precipitation, groundwater recharge, and streamflow. The sum of these two runoff responses quantitatively approximates the whole-catchment runoff response. Under wet conditions (here defined as antecedent water table depth  $\leq 1.66$  m), the first peak increases nonlinearly (particularly at precipitation intensities above 2 mm h<sup>-1</sup>) and the second peak becomes higher, narrower, and earlier with increasing precipitation intensity. Under dry conditions (here defined as antecedent water table depth > 1.66m), the first peak increases nonlinearly with precipitation intensity (particularly above 4 mm h<sup>-1</sup>), and groundwater recharge also responds to precipitation, but no clear second peak occurs regardless of precipitation intensity. The lack of a second peak under dry conditions plausibly arises from groundwater loss to evapotranspiration and from limited connectivity between groundwater and the stream, rather than from a lack of groundwater recharge. Almost no runoff response occurs at precipitation intensities below ~0.8 mm h<sup>-1</sup> under wet conditions and ~1.5 mm h<sup>-1</sup> under dry conditions. Above a precipitation-related threshold that initiates the first peak and a catchment wetness threshold that initiates the second peak, higher precipitation intensities amplify the first peak nonlinearly and trigger a larger and quicker second peak.

#### 1 Introduction

Despite decades of study, understanding runoff generation processes remains challenging. For example, isotopic and chemical tracers have shown that streamflow, even during peak discharges, is often comprised mostly of pre-event water ("old" water) stored in the catchment, rather than event water from recent precipitation ("new" water) (e.g., Alcaraz et al., 2024; Buttle, 1994; Camacho Suarez et al., 2015; Hoeg et al., 2000; Kirchner et al., 2000; Liu et al., 2004; Marx et al., 2021; Moore, 1989; Mosquera et al., 2016; Muñoz-Villers and McDonnell, 2012; Neal and Rosier, 1990; Sklash et al., 1976; Suecker et al., 2000). In other words, catchments store pre-event water in aquifers, soils or regolith (Cartwright and Morgenstern, 2018) for weeks, months, or even years, but then release it to streamflow within minutes, hours, or days following rainfall (Kirchner, 2003). Despite attempts to explain this prompt mobilization of old water with conceptual models such as transmissivity feedback (Bishop et al., 2004, 1990), kinematic waves (Beven, 1981), macropore flow (McDonnell, 1990), or fill and spill (Du et al., 2016; Tromp-van Meerveld and McDonnell, 2006), the physical mechanisms underlying this "old water paradox" are still poorly understood (Gabrielli et al., 2012; Kirchner, 2003; Kirchner et al., 2023; McDonnell and Beven, 2014).

- Some catchments exhibit double-peak (or bimodal) storm hydrographs, typically with a first peak that is almost simultaneous with precipitation, and a delayed second peak that produces more runoff (Anderson and Burt, 1977; Onda et al., 2001; Zillgens et al., 2007). Double-peak hydrographs have been observed in catchments spanning three orders of magnitude in area (e.g., nested catchments with areas of 0.07, 15.5, and 150 km² shown in Zillgens et al., 2007), with different land covers (e.g., forested (Haga et al., 2005; Martínez-Carreras et al., 2016) or intensively farmed (Birkinshaw, 2008)), in different geological settings (e.g., with 1–2 m freely drained brown earth soils above sandstone in a hillslope hollow and spur area (Anderson and Burt, 1977, 1978), or with 0–1 m soil mantle above shale or serpentinite rocks (Onda et al., 2001; Tsujimura et al., 1999), or with sandy soils in an inland valley (Masiyandima et al., 2003)), and in different climates (e.g., annual precipitation less than 700 mm (Cui et al., 2024) or over 2500 mm (Padilla et al., 2015)).
- Previous studies of double-peak hydrographs, including studies at the Weierbach catchment (Luxembourg) that is our focus here, have typically interpreted the two peaks as reflecting contributions of water with different ages from different landscape units of the catchment. The first runoff peak can be driven by precipitation falling into the stream, saturation-excess or infiltration-excess overland flow in near-stream areas or hillslope hollows, or lateral preferential flow through macropores along hillslopes (Anderson and Burt, 1978; Angermann et al., 2017; Birkinshaw, 2008; Cui et al., 2024; Glaser et al., 2016; Klaus et al., 2015; Rodriguez and Klaus, 2019). Thus the first runoff peak may be composed of both event water and pre-event water (Zillgens et al., 2007). The second peak is dominated by pre-event water and reflects subsurface processes involving shallow groundwater, deep subsurface flow through bedrock fissures, flow above the soil-bedrock interface, or hillslope throughflow (Anderson and Burt, 1978; Cui et al., 2024; Haga et al., 2005; Martínez-Carreras et al., 2016; Onda et al., 2001;

Schwab et al., 2017; Tsujimura et al., 1999; Wrede et al., 2015; Zillgens et al., 2007). The occurrence of double-peak hydrographs has been linked to thresholds of precipitation, antecedent wetness, and catchment storage that are site-specific and depend on catchment characteristics. For example, Zillgens et al. (2007) found delayed second peaks during storms with relatively high precipitation totals (> 40 mm), relatively low rainfall intensities (4–10 mm h<sup>-1</sup>), and wet conditions with high initial base flow. Martínez-Carreras et al. (2016) observed double-peak hydrographs only during wet conditions with catchment storage exceeding ~113 mm. And in two other studies, Haga et al. (2005) and Cui et al. (2024) found double peaks when the total storm volume plus the antecedent soil moisture index exceeded 135 mm and 200 mm, respectively. In an interesting historical example, double-peak hydrographs were commonly observed in the Schaefertal (Germany) in the 1970's, but became rare, only occurring in response to intense precipitation, after mining activities commenced underneath the catchment, leading to groundwater depletion by mine drainage (Graeff et al., 2009).

Compared to single-peak hydrographs, the delayed second peaks in double-peak hydrographs more clearly reflect the release of stored water to streamflow, so understanding the mechanisms that generate this second peak may shed light on the old-water paradox. Furthermore, understanding the processes underlying both peaks may be important because different flowpaths may transport different potential contaminants. However, a clear understanding remains incomplete, and predicting the occurrence of double-peak hydrographs remains difficult (Hissler et al., 2021; Martínez-Carreras et al., 2016). Moreover, most studies of double-peak runoff generation mechanisms rely on arbitrary assumptions to separate the hydrograph into baseflow and quickflow, or to isolate individual peaks and events from precipitation and runoff time series (Pelletier and Andréassian, 2020). Overlapping responses to fluctuating rainfall inputs can also make the second peak difficult to clearly define (Padilla et al., 2015).

Here we explore double-peak hydrograph generation by assimilating information from the entire catchment time series rather than individual runoff events, using ensemble rainfall-runoff analysis (ERRA; Kirchner, 2024a). This data-driven, model-independent, nonparametric approach eliminates the need to separate the hydrograph or identify individual runoff events, and allows us to quantify how double-peak runoff generation varies with precipitation intensity and antecedent wetness. We quantify the coupling between precipitation, groundwater recharge, and streamflow in the Weierbach catchment, including 1) how groundwater recharge and streamflow respond to precipitation, and how streamflow responds to groundwater recharge, over time; 2) how the individual effects of correlated inputs (precipitation and groundwater recharge) on runoff response differ from one another; and 3) how double-peak runoff response and each of its distinct peaks vary with changes in precipitation intensity and antecedent wetness conditions. Our case study in a forested headwater catchment suggests that the first spiky runoff response peak is dominated by precipitation directly entering the stream, while the delayed, lower, and broader second peak is primarily driven by precipitation which infiltrates to recharge groundwater, in turn triggering discharge from the

groundwater system to streamflow. Our results also demonstrate a precipitation threshold for initiating the first runoff response peak and an antecedent wetness threshold for initiating the second peak, above which higher precipitation intensities amplify the first runoff response peak nonlinearly and trigger a larger and quicker second runoff response.

#### 2 Materials and methods

## 2.1 Study site

The Weierbach experimental catchment (0.45 km²; Fig. 1) is a forested headwater catchment of the Attert River basin in Luxembourg, with annual average precipitation of ~804 mm and annual average streamflow of ~367 mm (2009–2019). The precipitation is rather evenly distributed throughout the year due to the semi-marine climate, whereas the base flow is lower from July to September due to higher evapotranspiration (~593 mm yr¹ during 2006–2014; Hissler et al., 2021; Pfister et al., 2017). Runoff response in this catchment is characterized by double-peak hydrographs under wet catchment conditions or during winter, and single-peak hydrographs under dry conditions or during summer (Martínez-Carreras et al., 2016; Schwab et al., 2017; Wrede et al., 2015).

The catchment ranges from 450 to 500 m in elevation on a sub-horizontal plateau cut by deep V-shaped valleys in the central Ardennes Massif (Hissler et al., 2021). The Devonian bedrock is mainly composed of schists, slate, phyllites, sandstones, and quartzites, and is covered by Pleistocene periglacial slope deposits (Juilleret et al., 2016; Pfister et al., 2017). The highly permeable cover beds are oriented parallel to the slope (Juilleret et al., 2011) and have two main layers: the "upper layer" from the surface to ~50 cm deep with a drainable porosity of 30%, and the "basal layer" from about 50 to 140 cm deep with a drainable porosity decreasing from 30% to 10% with increasing depth (Martínez-Carreras et al., 2016; Rodriguez and Klaus, 2019). Weathered and fractured bedrock starts from about 1.5 m depth and closes at about 5 m depth; deeper fresh bedrock is considered mostly impermeable (Gourdol et al., 2018; Rodriguez and Klaus, 2019).

Figure 1. Schematic map showing the location of the Weierbach catchment and monitoring sites for water table depth, soil water content, and streamflow measurements.

## 2.2 Hydrometric data

As the most instrumented and studied catchment in Luxembourg, the Weierbach catchment has been monitored using high-frequency hydro-meteorological measurements since 2009, including rainfall, soil water, groundwater, streamflow, isotopic composition, etc. Detailed descriptions of field sites, equipment, and data collection can be found in Hissler (2021), and the dataset is accessible at zenodo.org (Hissler et al., 2020).

Our analysis uses precipitation (P), water table depth (WTD), volumetric soil water content (VWC), and streamflow (Q) time series at the Weierbach catchment from September 2014 to December 2019 (Hissler et al., 2020). Precipitation was recorded at 10- and 15-min intervals from the Holtz rainfall monitoring station located 1 km from the catchment. Water table depth was recorded at 15-min and 1-hour intervals in 90-mm diameter plastic wells; we used the three wells with the most complete records covering the upper plateau, the middle of the hillslopes, and low hillslope positions in the catchment (GW2, GW3, and GW5; Fig. 1). Volumetric soil water content was measured at 10-cm, 20-cm, 40-cm, and 60-cm depth at five sites (Fig. 1) every 30 min using CS650 water content reflectometers (Campbell Scientific, Logan, Utah, USA). Discharge at the outlet was determined using water level measurements and rating curves.

To have a minimum uniform time interval for all variables at all sites throughout the study period, we aggregated the original measurements in the dataset into hourly time steps. To ensure that our ERRA analyses are based on time series that contain

consistent information and therefore represent consistent catchment behaviors, we used only complete records where P, WTD, and Q data are available at all sites (Fig. 2).

Groundwater recharge (GR) was calculated for each well by calculating the decrease in WTD (i.e., the increase in groundwater level) between each pair of hourly WTD measurements, multiplying by the drainable porosity, and then averaging the three wells to obtain the catchment-average GR. Drainable porosity was set to 10% for the depth range of the three groundwater sites (which have mean WTDs ranging from 1.3 to 2.7 m). This approach to estimating recharge from groundwater level fluctuations (also termed the water-table fluctuation method) is most valid when water recharges the water table at a greater rate than it leaves (Healy and Cook, 2002). When WTD increases (i.e., groundwater levels decline and estimates of GR are negative), recharge may still occur but is smaller than groundwater losses, and water table fluctuations under this circumstance will be more responsive to other factors such as evapotranspiration. To minimize the effect of these other factors on our estimates of GR, we have set all negative GR values to 0. Catchment-averaged variables (GR and WTD averaged over three wells, and VWC averaged over all probes at all depths) are used in the analyses presented here.

Figure 2. Overview of measured time series of precipitation (P), volumetric water content (VWC) for 4 depths (average of all available probes at each depth), water table depth (WTD) for 3 wells, and streamflow (Q) during the study period 2014–2019 at the Weierbach catchment. Catchment average groundwater recharge (GR) is calculated by averaging the GR from all wells. Only complete records with available measurements for variables P, GR, and Q at all sites were analyzed in this study.

#### 2.3 Ensemble rainfall-runoff analysis

We characterized and quantified the hydrological linkages between precipitation, groundwater recharge, and streamflow using ensemble rainfall-runoff analysis (ERRA; Kirchner, 2024a). ERRA is a data-driven, model-independent, nonparametric approach that quantifies nonlinear, nonstationary, and spatially heterogeneous hydrological behavior by combining least-squares deconvolution with de-mixing techniques and broken-stick regression. Readers are referred to Kirchner (2022) and Kirchner (2024a,b) for the relevant mathematical details, documentation, benchmark tests, proof-of-concept demonstrations, and calculation scripts. Here we only describe how we apply ERRA in our Weierbach analysis.

### 2.3.1 Runoff response distribution driven by precipitation (RRD<sub>P</sub>)

A simple rainfall-runoff system linking a single precipitation input (P) and a single streamflow output (Q) could potentially be approximated as a convolution with discrete time steps of length  $\Delta t$ :

$$Q_j = \sum_{k=0}^{m} RRD_{P,k} P_{j-k} \Delta t \tag{1}$$

where  $Q_j$  is streamflow at time step j,  $P_{j-k}$  is precipitation occurring k time steps earlier, RRD<sub>P,k</sub> is the impulse response of streamflow to precipitation at lag k, and m is the maximum lag being considered. The ensemble-averaged linear impulse response of streamflow to precipitation is termed the runoff response distribution (RRD), which is estimated by solving Eq. (1) via least-squares deconvolution of the streamflow time series by the precipitation time series in ERRA. ERRA also accounts for the effects of autoregressive moving-average noise, which is typically found in the residuals of Eq. (1) when it is applied to real-world hydrological time series. If Q and P are measured in the same units, the RRD has dimensions of time<sup>-1</sup>. The area under the RRD is not constrained to 1 and thus reflects mass imbalances due to, e.g., evapotranspiration losses or infiltration to deep groundwater.

#### 2.3.2 Groundwater recharge response distribution driven by precipitation (GRRDP)

Analogously to Sect. 2.3.1, an unsaturated zone system linking a single precipitation input (P) and a single (or spatially averaged) groundwater recharge output (GR) could potentially be approximated as a convolution with discrete time steps of length  $\Delta t$ :

$$GR_j = \sum_{k=0}^{m} GRRD_{P,k} P_{j-k} \Delta t$$
 (2)

where  $GR_j$  is groundwater recharge at time step j,  $P_{j-k}$  is precipitation occurring k time steps earlier,  $GRRD_{P,k}$  is the impulse response of groundwater recharge to precipitation at lag k, and m is the maximum lag being considered.

## 2.3.3 Runoff response distribution driven by groundwater recharge (RRD<sub>GR</sub>)

Analogously to Sect. 2.3.1, a saturated zone system linking a single (or spatially averaged) groundwater recharge input (GR) and a single streamflow output (Q) could potentially be approximated as a convolution with discrete time steps of length  $\Delta t$ :

$$Q_j = \sum_{k=0}^{m} RRD_{GR,k} GR_{j-k} \Delta t$$
 (3)

where  $Q_j$  is streamflow at time step j,  $GR_{j-k}$  is groundwater recharge occurring k time steps earlier,  $RRD_{GR,k}$  is the impulse response of streamflow to groundwater recharge at lag k, and m is the maximum lag being considered.

#### 2.3.4 Joint deconvolution and de-mixing of runoff responses to multiple drivers

The streamflow observed at the catchment outlet can be viewed as combining the effects of two distinct drivers. First, groundwater recharge (resulting from past precipitation inputs) will have lagged effects on streamflow. Second, precipitation inputs may also be directly reflected in streamflow response, without involving groundwater as an intermediary link. Each of these pathways can, at least in principle, be described by its own RRD, but streamflow will respond to both. Thus, separating their effects on streamflow requires a combination of deconvolution and de-mixing. We need to deconvolve the effects on streamflow from precipitation landing on the surface, and from groundwater recharge, while also de-mixing them from one another. This can be accomplished in ERRA by supplying both precipitation and groundwater recharge as inputs. ERRA will then attempt to de-convolve and de-mix the following statistical model:

$$Q_{j} = \sum_{k=0}^{m} \operatorname{partial} \operatorname{RRD}_{P,k} P_{j-k} \Delta t + \sum_{k=0}^{m} \operatorname{partial} \operatorname{RRD}_{GR,k} GR_{j-k} \Delta t$$
(4)

where partial RRD<sub>P,k</sub> and partial RRD<sub>GR,k</sub> are the partial runoff response distributions for precipitation bypassing groundwater and for groundwater recharge, respectively. ERRA un-scrambles the lagged effects of each input over time (deconvolution) and separates them from one another (de-mixing), at least up to the limitations of the available data.

#### 2.3.5 Nonlinear response functions

In real-world systems, streamflow often responds more-than-proportionally to changes in precipitation intensity. In a nonlinear rainfall-runoff system, in which the  $RRD_P$  at each lag is a function of the precipitation intensity, Eq. (1) becomes

$$Q_j = \sum_{k=0}^{m} P_{j-k} \operatorname{RRD}_{P,k}(P_{j-k}) \Delta t$$
 (5)

To characterize the functional relationship between precipitation intensity and streamflow response, a nonlinear response function (NRF) is defined as:

$$NRF_k(P_{i-k}) = P_{i-k} RRD_{P,k}(P_{i-k})$$
(6)

# 210 Combining Eqs. (5) and (6) yields

$$Q_j = \sum_{k=0}^m \text{NRF}_k(P_{j-k}) \ \Delta t \tag{7}$$

where  $Q_j$  is streamflow at time step j,  $P_{j-k}$  is precipitation occurring k time steps earlier, NRF $_k$  is the nonlinear response of streamflow to precipitation that falls at a rate  $P_{j-k}$  and lasts for a time step of  $\Delta t$ , m is the maximum lag being considered, and the parentheses indicate functional dependence rather than multiplication.

Similarly, in a nonlinear system linking precipitation and groundwater recharge, in which the  $GRRD_P$  at each lag is a function of the precipitation intensity, the NRF is expressed as

$$GR_j = \sum_{k=0}^m \text{NRF}_k(P_{j-k}) \ \Delta t \tag{8}$$

where  $GR_j$  is groundwater recharge at time step j,  $P_{j-k}$  is precipitation occurring k time steps earlier, NRF $_k$  is the nonlinear response of groundwater recharge to precipitation that falls at a rate  $P_{j-k}$  and lasts for a time step of  $\Delta t$ , and m is the maximum lag being considered. The NRFs in Eqs. (7) and (8) are approximated in ERRA by continuous piecewise-linear broken-stick functions of precipitation intensity (Fig. 3; see Kirchner 2022, 2024a for details). The NRF formally has units of mm  $h^{-2}$  (if P and Q are measured in mm  $h^{-1}$ ) because it expresses the incremental increase in streamflow that occurs in response to each time unit of precipitation at a given intensity. However, as explained in Kirchner (2024a), one can also consider the time step to be part of the definition of the NRF (e.g., an "hourly" NRF), in which case the units of the NRF become those of streamflow (e.g., mm  $h^{-1}$ ). We adopt this more intuitive interpretation for the NRFs presented here (keeping in mind the implicit time step of 1 h).

Figure 3. Schematic illustration of how a runoff response distribution (RRD) characterizes linear runoff response at a given lag k (a), and how a nonlinear response function (NRF) characterizes nonlinear runoff response (b). Grey points indicate how one time step of precipitation at a rate P alters discharge Q at a lag k. (ERRA statistically corrects these points for the overlapping effects of other precipitation inputs at other time lags, making them analogous to leverages in multiple regression.) If the runoff response is approximately linear, it can be approximated by the dashed line in (a), the slope of which is the RRD for that lag. If the runoff response is nonlinear, it can be approximated by a piecewise-linear relationship such as the dashed line in (b), connecting a series of knot points (open circles) at precipitation rates  $\kappa_0$ - $\kappa_4$ . Such relationships are functions of P and thus cannot be characterized by single values, like RRDs can. The NRF and the RRD have different dimensions because NRF estimates the effect of P on Q (the abscissa of (b), whereas the RRD estimates the slope of the relationship between P and P's effect on Q.

#### 3 Quantifying and de-mixing double-peak runoff response

#### 3.1 Streamflow and groundwater recharge response to single inputs

In this section, we use the methods outlined in Sect. 2.3.1, Sect. 2.3.2, and Sect. 2.3.3 to estimate response distributions (Fig. 4) that quantify the coupling between precipitation and streamflow, between precipitation and groundwater recharge, and between groundwater recharge and streamflow, respectively, averaged over the five years of record.

Figure 4a presents the runoff response distribution driven by precipitation (RRD<sub>P</sub>), quantified by using precipitation as the system input and streamflow as the system output in ERRA (see Eq. 1). The RRD<sub>P</sub> quantifies streamflow response per unit of precipitation over a range of lag times (here, up to a maximum lag of 240 hours = 10 days). Fig. 4a shows a double-peak streamflow response pattern. The first peak is a tall, narrow spike, occurring during the same hour that precipitation falls and the hour immediately following, with a peak height of  $0.0063 \pm 0.00006 \, h^{-1}$  (or 0.63% of precipitation per hour). The second peak is lower, broader, and significantly delayed, reaching a peak height of  $0.0020 \pm 0.0008 \, h^{-1}$  (or 0.2% of precipitation per hour) at a lag of ~37 hours. The integral under the RRD<sub>P</sub> yields an effective runoff coefficient of  $0.29 \pm 0.002$ , indicating that

about 29% of precipitation is eventually reflected in increased streamflow during the 240 hours after the rain falls. This 240-hour runoff coefficient is 63% of the long-term runoff coefficient (0.46, the ratio of average streamflow of 367 mm yr<sup>-1</sup> and average precipitation of 804 mm yr<sup>-1</sup>), suggesting that runoff responses shorter than 240 hours account for roughly two-thirds of streamflow in this catchment, with the remaining one-third comprising longer-term baseflow.

Figure 4b presents the groundwater recharge response distribution driven by precipitation (GRRD<sub>P</sub>), calculated by using precipitation as the system input and groundwater recharge as the system output in ERRA (see Eq. 2). The GRRD<sub>P</sub> quantifies groundwater recharge response to one unit of precipitation over a range of lag times, reflecting the transmission of hydrologic signals through the vadose zone. The GRRD<sub>P</sub> peaks 1 hour after precipitation falls, then declines to nearly zero within the next ~24 hours. The peak groundwater recharge response  $(0.088 \pm 0.002 \ h^{-1})$  is about 14 times the first peak of RRD<sub>P</sub>, and the integral of GRRD<sub>P</sub> is  $0.51 \pm 0.03$ , indicating that roughly half of precipitation is reflected in groundwater recharge within the first 240 hours after rain falls.

Figure 4c presents the runoff response distribution driven by groundwater recharge (RRD<sub>GR</sub>), estimated by using groundwater recharge as the system input and streamflow as the system output in ERRA (see Eq. 3). The RRD<sub>GR</sub> quantifies how streamflow responds to one unit of groundwater recharge over a range of lag times, reflecting the propagation of hydrologic signals through the saturated zone. The RRD<sub>GR</sub> exhibits a broad peak, similar to the second peak shown in the RRD<sub>F</sub> (Fig. 4a) but arriving somewhat earlier, with a peak lag of ~27 hours. The RRD<sub>GR</sub> also exhibits a sharp spike at near-zero lag; this may be an artifact caused by the strong short-lag relationships between precipitation and both streamflow (Fig. 4a) and groundwater recharge (Fig. 4b). In the following section, we further explore how this potential artifact can be reduced by jointly analyzing the effects of correlated precipitation and groundwater recharge on streamflow.

Figure 4. Response distributions estimated by ERRA at Weierbach. (a) Runoff response distribution driven by precipitation (RRD<sub>P</sub>). The runoff response consists of a tall brief spike, peaking at 0.0063 h<sup>-1</sup> (or 0.63% of precipitation per hour) within the first hour after rain falls, followed by a broader, lower second peak of 0.002 h<sup>-1</sup> (or 0.2% of precipitation per hour) at ~37 hours following rainfall. (b) Groundwater recharge response distribution driven by precipitation (GRRD<sub>P</sub>). The peak groundwater recharge response is much bigger (0.088 h<sup>-1</sup>) than the peak runoff response to precipitation (a), but decays to zero within ~24 h. (c) Runoff response distribution driven by groundwater recharge (RRD<sub>GR</sub>), exhibiting a broad peak at ~27h and a potentially artifactual spike at near-zero lag (see text). Standard errors are smaller than the plotting symbols.

#### 3.2 De-mixing streamflow responses to precipitation and groundwater recharge

The runoff response distributions presented in Sect. 3.1 describe streamflow response to either precipitation or groundwater recharge (in Figs. 4a and 4c, respectively), under the implicit assumption that each of these is the only driver of streamflow. The RRD<sub>P</sub>, estimated from deconvolving streamflow by precipitation alone, describes a whole-catchment system with precipitation as its sole input. The RRD<sub>GR</sub>, estimated from deconvolving streamflow by groundwater recharge alone, describes a saturated zone system with groundwater recharge as its sole input. In the real-world catchment, however, the streamflow observed at the catchment outlet reflects the overlapping effects of both precipitation and groundwater recharge, whose runoff responses may be differently lagged and dispersed but are overprinted on one another at the catchment outlet. Moreover, these two inputs are correlated, because the groundwater system is recharged by precipitation, while precipitation and groundwater recharge can both affect future streamflows. Therefore, we must separate the effects of precipitation and groundwater recharge on streamflow in order to accurately quantify each of them.

In this section, we use the combined deconvolution and de-mixing approach outlined in Sect. 2.3.4 to separate the overlapping effects of precipitation and groundwater recharge on streamflow, by using them both as joint inputs to ERRA. The de-mixed runoff response distribution driven by precipitation (partial RRD<sub>P</sub>) and de-mixed runoff response distribution driven by groundwater recharge (partial RRD<sub>GR</sub>) are shown in Fig. 5.

The de-mixed runoff response distribution driven by precipitation ( $^{partial}RRD_P$ ) quantifies runoff response to precipitation when groundwater recharge is also accounted for; in other words, it quantifies how precipitation affects streamflow directly, without groundwater recharge as an intermediary. In contrast to the  $RRD_P$  (Fig. 4a) with its spiky first peak and broader second peak, the  $^{partial}RRD_P$  shown in Fig. 5 has no substantial second peak. Instead, the  $^{partial}RRD_P$  peaks at  $0.0061 \pm 0.00006 \, h^{-1}$  during the same hour that precipitation falls, then rapidly declines within ~12 hours to stabilize near zero. The peaks in the  $^{partial}RRD_P$  and  $^{partial}RRD_P$  occur at similar lags and have similar magnitudes ( $0.0061 \, h^{-1}$  versus  $0.0063 \, h^{-1}$ ), implying that the initial peak in the  $^{partial}RRD_P$  is driven primarily by the direct effects of precipitation on streamflow.

The de-mixed runoff response distribution driven by groundwater recharge (partial RRD<sub>GR</sub>) quantifies how runoff responds to

groundwater recharge when precipitation is also accounted for; in other words, it quantifies how groundwater recharge affects streamflow, while correcting for the potentially confounding direct effects of precipitation on streamflow. The partial RRD<sub>GR</sub> in Fig. 5 has a similar broad and delayed peak as in the RRD<sub>P</sub> (Fig. 4a), suggesting that groundwater recharge is the dominant source of the second broad peak in streamflow. Compared to the RRD<sub>GR</sub> (Fig. 4c), the partial RRD<sub>GR</sub> has a smaller spike at the

first point  $(0.004 \pm 0.0002 \text{ in } ^{partial} RRD_{GR} \text{ versus } 0.006 \pm 0.0002 \text{ in } RRD_{GR})$ . This difference illustrates that the  $RRD_{GR}$  derived by coupling streamflow with the single groundwater recharge input can be distorted due to the strong correlation and short-lag

response between precipitation and groundwater recharge. The remaining short-lag spike in  $partialRRD_{GR}$  may indicate that this distortion cannot be completely eliminated by the de-mixing approach of Sect. 2.3.4. Alternatively, the remaining short-lag spike in  $partialRRD_{GR}$  could potentially be real, reflecting rapid runoff effects of groundwater recharge in the near-stream zone. Unfortunately, we lack the necessary data to test either of these hypotheses.

Readers will note that the direct runoff response to precipitation ( $^{partial}RRD_P$ ) in Fig. 5 occasionally dips below zero at long lags. This is the expected result of statistical noise, given that the direct effect of precipitation on streamflow decays to nearly zero within the first ~12 hours; thus the longer lags can be expected to be dominated by statistical fluctuations. Readers will also note that the runoff response to groundwater recharge ( $^{partial}RRD_{GR}$ ) in Fig. 5 does not converge to zero, even after 240 hours. It is unknown whether this is a statistical artifact or a reflection of long groundwater lags. The integral under the  $^{partial}RRD_{GR}$  is 0.617  $\pm$  0.003, suggesting that ~40% of recharge could potentially remain to be discharged at longer lags. Such an estimate is inherently uncertain, however, because it does not account for evapotranspiration losses from groundwater (which would reduce the amount of recharge remaining for later discharge), and does not account for the inherent underestimation of recharge in the water table fluctuation method (which would imply more recharge remaining for later discharge).

Figure 5. De-mixed runoff response distributions, estimated by deconvolving and de-mixing the effects of both precipitation and groundwater recharge on streamflow. The de-mixed runoff response distribution driven by precipitation (Partial RRDP, solid symbols) is different from the total runoff response distribution driven by precipitation alone (RRDP, shown in Fig. 4a); they have similar initial peaks, but Partial RRDP lacks the second peak that dominates RRDP. The de-mixed runoff response distribution driven by groundwater recharge (Partial RRDGR, open symbols) has a smaller short-term spike than the total runoff response distribution driven by groundwater recharge alone does (RRDGR, shown in Fig. 4c), although they have similar broad delayed peaks. This deconvolution and de-mixing analysis suggests that the direct streamflow response to precipitation (solid symbols) differs greatly from the streamflow response to groundwater recharge (open symbols). Error bars show one standard error, where this is larger than the plotting symbols.

The individual runoff responses to precipitation and groundwater recharge presented in this section roughly align with the patterns of the two peaks in the total streamflow response driven by precipitation alone (RRD<sub>P</sub> in Fig. 4a). This suggests that precipitation affects streamflow both directly, and indirectly via groundwater recharge, with each process dominating one of the peaks. One apparent discrepancy, however, is that the second peak in the RRD<sub>P</sub> in Fig. 4a occurs at a lag of ~37 hours, roughly 10 hours later than the peak in the  $^{partial}$ RRD<sub>GR</sub> at ~26 hours. As we will see in Sect. 3.3 below, this difference in lag times can be explained by taking account of the full spectrum of lag times for precipitation to become groundwater recharge.

#### 3.3 Double-peak runoff generation resulting from near-surface and groundwater-mediated pathways

## 3.3.1 Two-pathway hypothesis

The response distributions presented in Sects. 3.1 and 3.2 imply that precipitation influences streamflow via two main pathways:

- (1) precipitation directly influences streamflow, leading to the spiky first peak in the streamflow response. This pathway characterizes the direct effect of precipitation falling directly into the stream or onto near-stream saturated areas. We refer to it as the "near-surface pathway" hereafter;
- (2) precipitation recharges groundwater, which then contributes to streamflow. This pathway presumably dominates the lower and broader second peak in the streamflow response. We refer to this as the "groundwater-mediated pathway" hereafter.
- We can test this runoff generation hypothesis (Fig. 6) by exploring whether the total runoff response to precipitation (Fig. 4a) can be quantitatively explained by combining the individual streamflow components resulting from the two pathways described above.
  - The total runoff response distribution driven by precipitation (RRD<sub>P</sub>) is calculated by deconvolving streamflow by precipitation alone. Hydrologically, RRD<sub>P</sub> describes the average behavior of all pathways linking precipitation P and streamflow Q. Mathematically, it is the convolution kernel of the whole rainfall-runoff system. The convolution of the whole-catchment rainfall-runoff system is denoted as

$$P * RRD_P = 0 (9)$$

where the star symbol denotes convolution.

The near-surface pathway of runoff generation (precipitation → runoff) can be characterized by the partial runoff response distribution driven by precipitation (partial RRDp) resulting from deconvolving and de-mixing the joint effects of precipitation and groundwater recharge on streamflow. The partial RRDp describes the direct hydrological effect of precipitation on streamflow,

by factoring out the effects of groundwater recharge, i.e., it quantifies the response behavior of the near-surface pathway that directly links precipitation to streamflow. Mathematically, the  $^{partial}RRD_P$  is the convolution kernel of the system connecting precipitation P to the streamflow component that results from the near-surface pathway (here denoted  $Q_1$ ). Therefore the near-surface pathway can be expressed as

$$P * \text{partial} RRD_{P} = Q_{1} \tag{10}$$

The groundwater-mediated pathway of runoff generation (precipitation $\rightarrow$ groundwater $\rightarrow$ streamflow) assumes a causal chain linking precipitation P to groundwater recharge (GR) and the resulting streamflow component (here denoted  $Q_2$ ). The vadose zone system (precipitation $\rightarrow$ groundwater) can be characterized by the total groundwater recharge response distribution driven by precipitation (GRRD<sub>P</sub>), formed by deconvolving groundwater recharge by precipitation. The GRRD<sub>P</sub> is the convolution kernel of the precipitation $\rightarrow$ groundwater recharge system, denoted as

$$P * GRRD_{p} = GR \tag{11}$$

The saturated groundwater system (groundwater $\rightarrow$ streamflow) can be characterized by the partial runoff response distribution driven by groundwater recharge (partialRRD<sub>GR</sub>) estimated by deconvolving and de-mixing the joint effects of precipitation and groundwater recharge on streamflow. The partialRRD<sub>GR</sub> describes the effects of groundwater recharge on streamflow by factoring out the direct effects of precipitation on streamflow. It is the convolution kernel of the system linking groundwater recharge and the streamflow component  $Q_2$ , denoted as

$$GR * \text{partial} RRD_{GR} = Q_2$$
 (12)

Combining Eqs. (11) and (12) yields

385

395

$$(P * GRRD_P) * partial RRD_{GR} = Q_2$$
 (13)

390 By the associative property of convolution, Eq. (13) becomes

$$P * (GRRD_P * partialRRD_{GR}) = Q_2$$
 (14)

Hydrologically, Eq. (14) expresses a convolution chain representing the groundwater-mediated pathway linking precipitation to the corresponding streamflow component  $Q_2$ , where precipitation initially infiltrates and recharges groundwater, followed by discharge from groundwater to streamflow. Mathematically, convolving the convolution kernels of the vadose zone and the saturated zone (as in Eq. 14) should yield a good approximation for the groundwater-mediated pathway only if there is actually a causal chain connecting precipitation to groundwater recharge and then to streamflow.

If the streamflow Q at the catchment outlet mainly consists of the streamflow component  $Q_1$  resulting from the near-surface pathway and the streamflow component  $Q_2$  resulting from the groundwater-mediated pathway, then Q should be closely approximated by the sum of these two components:

$$Q = Q_1 + Q_2 (15)$$

Combining Eqs. (9), (10), (14) and (15) yields

$$P * RRD_P = P * partial RRD_P + P * (GRRD_P * partial RRD_{GR})$$
 (16)

By the distributive property of convolution, Eq. (16) becomes

$$P * RRD_P = P * (partial RRD_P + GRRD_P * partial RRD_{GR})$$
 (17)

which is equal to

$$RRD_{P} = {}^{partial}RRD_{P} + GRRD_{P} * {}^{partial}RRD_{GR}$$
(18)

410 Therefore, the hypothesis outlined above can be tested by exploring whether Eq. (18) holds.

Figure 6. Diagram illustrating two potentially dominant pathways contributing to double-peak runoff response. (a) Runoff response distribution (RRD<sub>P</sub>), a convolution kernel linking precipitation and streamflow of the whole rainfall-runoff system. (b) The direct effect of precipitation on streamflow through the near-surface pathway (on the blue background) is represented by the partial runoff response distribution driven by precipitation ( $^{\text{partial}}$ RRD<sub>P</sub>), and the effect of water on streamflow through the groundwater-mediated pathway (on the yellow background) is represented by the convolution of the convolution kernel of the vadose zone system linking precipitation and groundwater recharge (GRRD<sub>P</sub>) and the de-mixed partial runoff response distribution driven by groundwater recharge ( $^{\text{partial}}$ RRD<sub>GR</sub>) in the saturated zone system. If streamflow is generated by a combination of the direct effect of precipitation (near-surface pathway) and a causation chain linking precipitation to the groundwater system and then to streamflow (groundwater-mediated pathway), then the convolution kernel of the whole-catchment rainfall-runoff system (RRD<sub>P</sub>) should be approximated by the sum of the convolution kernels of the near-surface pathway ( $^{\text{partial}}$ RRD<sub>P</sub>) and the groundwater-mediated pathway (GRRD<sub>P</sub> \*  $^{\text{partial}}$ RRD<sub>GR</sub>).

## 425 **3.3.2** Result for two-pathway hypothesis

430

435

Figure 7a shows good agreement between both sides of Eq. (18). The total runoff response distribution driven by precipitation for the whole-catchment rainfall-runoff system (RRD<sub>P</sub>, shown in dark blue in Figs. 7a–c and Fig. 4a) is almost exactly reproduced by the sum of the runoff response distributions sourced from the near-surface pathway and groundwater-mediated pathway ( $^{partial}RRD_P + GRRD_P * ^{partial}RRD_{GR}$ , shown in green in Fig. 7a). The sum of runoff response distributions is overall slightly bigger than the RRD<sub>P</sub> (the integrals under the dark blue and green curves in Fig. 7a are 0.29 and 0.33, respectively).

Figures 7b and 7c further show that the runoff response distributions for each pathway are plausible sources for each of the runoff response peaks in the RRD<sub>P</sub>. The first spiky peak in RRD<sub>P</sub> is well matched by the runoff response distribution resulting from the near-surface pathway (partial RRD<sub>P</sub>, shown light blue in Fig. 7b and Fig. 5). The second broad peak with the long recession process is well represented by the runoff response distribution resulting from the groundwater-mediated pathway (GRRD<sub>P</sub> \* partial RRD<sub>GR</sub>, shown in orange in Fig. 7c). Runoff response from each pathway effectively captures the shape, magnitude, and timing of each peak in the RRD<sub>P</sub>.

Figures 7b and 7c also reject two alternative hypotheses. The disconnect between the two curves in Figure 7b rejects the hypothesis that the near-surface pathway alone can explain the total rainfall-runoff response (because it doesn't explain the second peak). Similarly, the disconnect between the two curves in Figure 7c rejects the hypothesis that the groundwater-mediated pathway alone can explain for the total rainfall-runoff response (because it doesn't explain the first peak).

The evidence in Fig. 7 strongly suggests that the Weierbach catchment's behavior is consistent with the runoff generation hypothesis outlined in Sect. 3.3.1. Part of the precipitation falling onto the catchment influences streamflow directly through near-surface processes and dominates the first large runoff response peak (Fig. 7b), which arrives within the first hour and declines rapidly within 3 hours. Another part of the precipitation infiltrates to recharge the groundwater, triggering groundwater discharge to streamflow and thus generating the second runoff response peak (Fig. 7c), which reaches about 1/3 the height of the first peak within about 48 hours and then gradually decays over the following ~200 hours.

Figure 7. Testing the double-peak runoff generation hypothesis. (a) Comparison between the total runoff response to precipitation of the whole-catchment rainfall-runoff system (RRDP, dark blue) and the sum of the runoff responses through the near-surface pathway and the groundwater pathway ( $^{partial}RRD_P + GRD_P * ^{partial}RRD_GR$ , green). (b) Comparison between the first peak in RRDP (dark blue) and the direct runoff response to precipitation  $^{partial}RRD_P$  (light blue). (c) Comparison between the second peak in RRDP and the groundwater-mediated runoff response  $GRD_P * ^{partial}RRD_GR$  (orange). The good match implies that the double-peak runoff response at Weierbach can be explained by the combined effects of near-surface runoff, which dominates the sharp first peak, and groundwater-mediated runoff, which dominates the lower and broader second peak.

#### 4 Quantifying nonlinearity and nonstationarity in double-peak runoff response

Section 3 demonstrated how precipitation shapes streamflow at the catchment outlet by near-surface and groundwater-mediated pathways. The analysis presented above characterizes these effects in an ensemble-averaged sense, but in practice they may vary depending on precipitation intensity and ambient catchment conditions. For example, runoff may respond more-than-proportionally to changes in precipitation intensity (nonlinearity), or may respond differently depending on the catchment wetness status when the rain falls (nonstationarity). This naturally raises the question of whether the runoff responses generated by the near-surface and groundwater-mediated pathways exhibit different degrees of nonlinearity and nonstationarity.

Here we quantify the nonlinear and nonstationary response behaviors of the Weierbach catchment using the methods outlined in Sects. 2.3.4 and 2.3.5 to measure how runoff responds to different precipitation intensities and antecedent wetness conditions. To reduce the uncertainty in the runoff response at long lags (arising from the weakness of the signals in the long recession tail observed in Sect. 3), we use ERRA's broken-stick approach, which estimates runoff response over wider lag ranges at longer lag times, rather than estimating runoff response at each individual hourly lag (see Sect. 5 of Kirchner, 2024a for details). The resulting runoff response distributions analyze the same 240-hour lag time scale, closely follow each hour's runoff response at short lags (where signals are strong), and closely follow the average runoff response at long lags (where signals are weak and individual hourly lag estimates would be noisy).

## **4.1** Antecedent wetness controls on runoff response (nonstationarity)

470

485

Using the method outlined in Sect. 2.3.4, we compare RRDs between different antecedent wetness categories to quantify how antecedent wetness influences the runoff response of the whole-catchment rainfall-runoff system (i.e., nonstationary runoff response). As a proxy for antecedent wetness at the catchment scale, we use the antecedent catchment-averaged water table depth (antWTD) measured 6 hours before precipitation falls. We separated the antecedent WTD into 3 ranges: shallower than 1.30 m (the shallowest 5% of WTD values), 1.30–1.66 m (the 5<sup>th</sup>–30<sup>th</sup> percentiles of WTD values), and deeper than 1.66 m (the deepest 70% of WTD values).

Figure 8 shows runoff response distributions for these three antecedent WTD ranges, with shallower WTD conditions (i.e., wetter catchment conditions) shown in darker blue. When the water table is deep (>1.66 m, the driest condition among the 3 antWTD ranges, shown in light blue in Fig. 8), the near-surface pathway generates a substantial peak response within the first hour after precipitation falls, but the groundwater-mediated pathway generates negligible runoff response. The same unit of precipitation falling when the catchment is wetter (i.e., its water table is shallower; medium blue and dark blue symbols in Fig. 8) triggers a larger first peak in runoff response within the first hour after precipitation falls. It also generates a second peak that grows higher, narrower, and earlier as antecedent wetness increases (i.e., as antecedent water table depth decreases).

The physical mechanisms underlying these patterns of response remain speculative. Wetter conditions may expand near-stream zones that are close to saturation, thus enhancing the direct effect of precipitation on streamflow via the near-surface pathway. Wetter conditions may also improve subsurface permeability and connectivity, thus enhancing and accelerating infiltration to the water table. Shallower water tables may also intersect with higher-permeability layers of the subsurface (the transmissivity feedback hypothesis (Bishop et al., 2004, 1990)).

Figure 8. Nonstationary runoff response distributions driven by precipitation inputs alone (RRD<sub>P</sub>) under different antecedent wetness conditions (represented by antecedent water table depth (antWTD) 6 hours before precipitation falls). Inset figure shows the first 5 hours of runoff response (corresponding to the thin gray-shaded area) in greater detail. When antecedent wetness is low (antWTD is deeper), precipitation generates a single-peak runoff response (light blue symbols) via the near-surface pathway, with no clear second peak. When the catchment is wetter before precipitation falls (shown in medium and dark blue), the same precipitation generates a second peak by triggering water release from the groundwater-mediated pathway. Wetter antecedent conditions enhance and accelerate water release, reflected in a higher, narrower, and earlier second peak (as well as a higher first peak). Error bars indicate one standard error, where this is smaller than the plotting symbols.

## 4.2 Precipitation intensity and antecedent wetness controls on runoff response (nonlinearity and nonstationarity)

Here we jointly analyze the influence of precipitation intensity and antecedent wetness on runoff response using the method outlined in Sect. 2.3.4 and Sect. 2.3.5. To quantify how runoff responds to different precipitation intensities under wet vs. dry ambient conditions (i.e., both nonlinear and nonstationary runoff response), we estimate NRFs for both "dry conditions" (the driest 70% of antecedent water table depths, antWTD >1.66 m, which exhibited a single-peak runoff response in Sect. 4.1), and "wet conditions" (the wettest 30% of antecedent WTD values, antWTD  $\leq$ 1.66 m, which exhibited a double-peak runoff response in Sect. 4.1).

ERRA can jointly analyze how runoff responds to different ranges of precipitation intensity and antecedent wetness, while accounting for their overlapping effects through time (see Kirchner, 2024a for details). At Weierbach, the highest precipitation intensities occur in summer, when the catchment is usually relatively dry; conversely, the range of precipitation intensities is narrower in the winter, when the catchment is wetter. Therefore we analyzed different precipitation intensity

intervals for wet and dry catchment conditions, instead of applying the same intervals to both. For each antecedent wetness condition, we specified 4 precipitation intensity intervals that divide the full range of precipitation intensities (under those wetness conditions) as evenly as possible, with the constraint that each interval must contain at least 60 valid data points for analysis. The resulting nonlinear response functions (NRFs, see Eq. 7) quantify how runoff responds to one time step (1 hour) of precipitation falling within the specified ranges of intensity and antecedent wetness. The comparison of NRF curves within each antecedent wetness category reflects only nonlinear runoff response, whereas the comparison of NRFs between wet and dry categories jointly reflects both nonlinear and nonstationary runoff response.

Under dry antecedent wetness conditions (Fig. 9a), runoff response exhibits a single-peak pattern, without a clear second runoff response peak, across all precipitation intensity ranges. The peak values of this runoff response increase nonlinearly with precipitation intensity, particularly above precipitation intensities of about 4 mm h<sup>-1</sup> (Fig. 9c).

If precipitation falls when antecedent wetness is high (Fig. 9b), runoff response exhibits a second peak that becomes higher, narrower, and earlier with increasing precipitation intensity. The first runoff response peak grows nonlinearly with precipitation intensity, particularly above rainfall rates of roughly 2 mm h<sup>-1</sup>, while the second runoff response peak grows almost linearly with precipitation intensity (Fig. 9c). The first peak increases somewhat more steeply than the second peak does, and increases more steeply under wet conditions than under dry conditions.

Figure 9 illustrates the joint dependence of runoff response on precipitation intensity and antecedent wetness. Higher precipitation intensities amplify the first runoff response peak but do not substantially change its timing under both wet and dry conditions. In contrast, higher precipitation intensities alter both the timing and the magnitude of the second runoff response peak, but only under wet antecedent conditions. The lowest precipitation intensity yields very weak runoff response regardless of antecedent wetness conditions. Conversely, under dry antecedent wetness conditions, even intense precipitation does not trigger a second runoff peak, implying that the groundwater system cannot transmit precipitation signals to streamflow when the catchment is not wet enough. These results suggest a precipitation intensity threshold in the initiation of the first runoff response peak, and a catchment wetness threshold in the initiation of the second runoff response peak, above which the effects of increasing precipitation intensity on both the first and the second peaks become pronounced. These results also support the hypothesis that precipitation contributes directly to the first peak through the near-surface pathway, and to the second peak via the groundwater-mediated pathway.

Figure 9. Nonlinear and nonstationary runoff responses quantified by nonlinear response functions (NRFs) under (a) dry and (b) wet antecedent conditions. Inset in (a) shows the first 7 hours of runoff response (corresponding to the thin gray-shaded area) in greater detail. (c) Peak runoff responses (i.e., the peaks of the curves in (a) and (b)) as a function of precipitation intensity under wet and dry antecedent conditions. At the Weierbach catchment, precipitation intensities are more variable in the summer, when ambient conditions are drier. Under dry antecedent conditions (a), runoff response exhibits only a single peak, even at high precipitation intensities. The second peak only emerges under wet antecedent conditions (b), and is higher, narrower, and earlier at higher precipitation intensities. These results suggest a precipitation intensity threshold for initiation of the first peak and an antecedent wetness threshold for initiation of the second peak. Error bars indicate one standard error.

## **5 Discussion**

# 5.1 Nonlinear and nonstationary double-peak runoff response to precipitation

Our results provide a new quantitative view that complements previous explorations of the double-peak runoff response at Weierbach. Previous studies at the Weierbach catchment suggest that the first peak mainly consists of event water (Martínez-Carreras et al., 2015) from rain falling directly into the stream, runoff generated in the riparian zone (Glaser et al., 2016; Klaus et al., 2015; Rodriguez and Klaus, 2019) and lateral preferential flow (Angermann et al., 2017). The second peak has been shown to only occur after the exceedance of a catchment storage threshold (~113 mm; Martínez-Carreras et al., 2016) and its timing is inconsistent with the activation of preferential flow paths in the shallow subsurface (Angermann et al., 2017). The second peak has been inferred to be mainly composed of pre-event water released from groundwater storage (Martínez-Carreras et al., 2015; Schwab et al., 2017; Wrede et al., 2015).

Surface saturation and stream network dynamics have been shown to relate to discharge to varying degrees. Thermal IR mapping of riparian areas at Weierbach has shown that surface saturation is related to discharge by power-law relationships (Antonelli et al., 2020a) that eventually mirror the degree of connectivity between saturated surfaces and the subsurface system across different riparian areas. However, these relationships varied across the Weierbach catchment, mainly associated with the location of the riparian areas and possible influences of local riparian morphology on surface saturation dynamics. Stream network extension and retraction, as expressions of the general wetness state of the catchment, have been shown to relate to groundwater fluctuations and changes in catchment storage (Antonelli et al., 2020b). However, in contrast to the dynamic expansion and contraction of near-stream saturated areas, stream network extension and retraction were found not to be very responsive to changes in discharge at the Weierbach's outlet. In other words, at Weierbach, perennial springs 'anchor' the channel heads in specific locations for the most part.

Our analysis adds to these previous studies by quantifying the coupling between precipitation, groundwater dynamics, and streamflow, and by exploring how these linkages vary with antecedent wetness and precipitation intensity. We show that the whole-catchment runoff response (RRD<sub>P</sub>, Fig. 4a) can be quantitatively represented as the sum of two components (Fig. 7). The first component is a rapid direct response to precipitation inputs (partial RRD<sub>P</sub>, Fig. 7b), and the second, slower component comprises the response of groundwater recharge to precipitation, convolved with the response of streamflow to groundwater recharge (GRRD<sub>P</sub> \* partial RRD<sub>GR</sub>, Fig. 7c). Thus our analysis is consistent with the view that the first runoff response peak results from a near-surface pathway directly linking precipitation and streamflow, while the delayed second peak is dominated by a causation chain in which precipitation infiltrates to recharge groundwater, which in turn triggers groundwater discharge to streamflow. By de-mixing the effects of these two pathways on streamflow, ERRA allows them both to be quantified.

Previous work at Weierbach has observed single-peak hydrographs under dry catchment conditions or during summer, and double-peak hydrographs under wet conditions or during winter (Martínez-Carreras et al., 2016; Schwab et al., 2017; Wrede et al., 2015). Our analysis refines these observations by quantifying how runoff responds to differences in precipitation and antecedent wetness (Figs. 8–9). The first runoff peak is found under both wet and dry antecedent conditions, but is more sensitive to precipitation intensity under wet conditions. Double-peak hydrographs emerge only under wet antecedent conditions, with the second peak becoming higher, narrower and earlier at higher precipitation intensities. This suggests a wetness-related threshold to initiate the second runoff response peak, above which higher precipitation intensities trigger more water release from the catchment more quickly, potentially through increases in subsurface connectivity. These discussed behaviors are specific to runoff generation mechanisms in the Weierbach catchment, based on the data we actually have, and have actually analyzed. Although the inferred runoff mechanisms are not amenable to generalization yet, the methods and hypotheses presented here may provide useful insights for explorations in other catchments, and in inter-catchment comparison studies.

The lack of a second peak could hypothetically arise either from a lack of recharge, from depletion of groundwater by evapotranspiration, or from a lack of connectivity between groundwater and the stream when water tables are low. In Fig. 10, we compare nonlinear groundwater recharge response to precipitation (see Eq. 8) under wet and dry conditions. Fig. 10 shows that even under dry conditions, groundwater recharge responds to precipitation, although at only about half the rate as during wet conditions (average groundwater recharge rates are  $0.055 \pm 0.001$  and  $0.104 \pm 0.003$  mm h<sup>-1</sup> in dry and wet conditions, respectively). But recharge under wet conditions is more effectively translated into discharge: average streamflow under wet conditions is 0.112 mm h<sup>-1</sup>, or ~100% of mean groundwater recharge, whereas average streamflow under dry conditions is 0.009 mm h<sup>-1</sup>, or ~16% of mean groundwater recharge. Considered together, these observations suggest that the lack of a second peak during dry conditions cannot be attributed to a lack of groundwater recharge, but more plausibly may arise from groundwater losses to evapotranspiration and from limited connectivity between groundwater and the stream.

Figure 10. Peak height of nonlinear response functions (NRFs), showing peak groundwater recharge response as a function of precipitation intensity under wet and dry conditions. Error bars indicate one standard error.

#### 5.2 Comparison of different proxies for catchment antecedent wetness conditions

Catchment antecedent wetness conditions can be described by a range of proxy measurements. For example, available proxies at Weierbach include antecedent water table depth (antWTD), antecedent volumetric water content (antVWC), and antecedent streamflow (antQ). Soil moisture measurements here only reflect the wetness state of the upper 60 cm of the subsurface (Fig. 2). The interpretation of antecedent streamflow necessarily depends on whether that streamflow results from the first peak (which is driven primarily by precipitation intensity) or the second peak (which is driven primarily by catchment wetness, and more specifically groundwater). Therefore, considering the evident role of groundwater in generating the second peak, we used antWTD as a proxy for antecedent wetness in our assessment of nonstationary runoff response to precipitation in Sect. 4.

In this section, we compare antWTD with antVWC and antQ as alternative proxies for catchment-scale antecedent wetness in runoff response analyses. WTD is averaged among three available wells, reflecting the catchment wetness status at depths of ~0.6–3 m. VWC is averaged among all available probes at all depths, reflecting the catchment wetness status of the unsaturated zone in the upper 60 cm. Streamflow itself reflects the integrated catchment wetness status. We tested each of these antecedent wetness proxies crossed with four categories of antecedent time lag (antWTD, antVWC, and antQ, measured 1, 6, 12, and 24 hours before precipitation falls). For each proxy, runoff response distributions are estimated for three ranges of wetness, delimited by the 5<sup>th</sup> and 30<sup>th</sup> percentiles of the WTD distribution (equivalent to the 95<sup>th</sup> and 70<sup>th</sup> percentiles of the groundwater level distribution), the 60<sup>th</sup> and 95<sup>th</sup> percentiles of the VWC distribution, and the 30<sup>th</sup> and 60<sup>th</sup> percentiles of the streamflow distribution.

Each panel in Fig. 11 presents runoff response distributions under three levels of antecedent wetness. The nonstationary runoff responses shown in Fig. 11 align with those shown in Sect. 4.1; the first peak runoff response is higher and the delayed second peak runoff response is higher and quicker under wetter antecedent conditions (shown in dark blue in each panel). However, when antecedent streamflow is the proxy for antecedent wetness (right column in Fig. 11), the second peak is nearly the same between the driest range (antQ 

Figure 11. Comparison of antecedent water table depth (antWTD, left column), antecedent volumetric water content (antVWC, middle column), and antecedent streamflow (antQ, right column) as proxies for catchment antecedent wetness in analyzing runoff responses to precipitation under wet and dry conditions.

#### **6 Conclusions**

We used ensemble rainfall-runoff analysis (ERRA), a data-driven, model-independent, nonparametric deconvolution and demixing approach, to characterize and quantify double-peak runoff generation at Weierbach, a forested headwater catchment in Luxembourg. Jointly analyzing precipitation and groundwater recharge as combined inputs in ERRA effectively separates and quantifies their individual effects on streamflow. The direct effect of precipitation on streamflow through the near-surface pathway dominates the first runoff response peak (Fig. 7b), which is high and sharp, peaking within the first hour after precipitation falls and rapidly declining to nearly zero after a few hours. Precipitation that infiltrates to groundwater, and thus triggers groundwater release to streamflow, dominates the second runoff response peak (Fig. 7c). Relative to the first peak, this second peak is later (peaking at about 1.5 days after precipitation falls), lower (about 1/3 the height of the first peak), and broader (declining to nearly zero after ~10 days).

Quantification of both nonlinear and nonstationary runoff response to precipitation shows that the first runoff response peak increases nonlinearly with precipitation intensity, particularly above rainfall rates of about 4 mm h<sup>-1</sup> under dry conditions and about 2 mm h<sup>-1</sup> under wet conditions (Fig. 9). Nearly no runoff response occurs at the lowest precipitation intensity regardless of antecedent wetness conditions, and no clear second delayed runoff response peak occurs when precipitation falls under dry conditions regardless of precipitation intensity. These observations suggest a precipitation-related threshold to initiate the first runoff response peak and a catchment wetness threshold to initiate the second peak, after which higher precipitation intensities amplify the first runoff response and trigger a larger and quicker second runoff response.

Quantifying the coupling between precipitation and groundwater recharge under wet and dry conditions (Fig. 10) shows that groundwater recharge responds to precipitation even when the catchment is dry (at about half the rate under wet conditions), but is more effectively translated into streamflow when the catchment is wet. These results suggest that the lack of a second runoff response peak under dry conditions may primarily arise from groundwater depletion due to evapotranspiration and/or from limited connectivity between groundwater and the stream, instead of from a lack of groundwater recharge.

#### Code and data availability

The Weierbach hydrological database is available at https://doi.org/10.5281/zenodo.4537700 (Hissler et al., 2020). The Ensemble Rainfall-Runoff Analysis (ERRA) script, along with introductory documentation for users, is available at https://doi.org/10.16904/envidat.529 (Kirchner, 2024b); our analysis is based on ERRA version 1.05.

#### **Author contributions**

HG: conceptualization, data curation, formal analysis, methodology, software, validation, visualization, writing – original draft preparation, writing – review and editing. LP: investigation, resources, writing – review and editing. JWK: conceptualization, formal analysis, methodology, software, supervision, validation, visualization, writing – review and editing.

## 690 Competing interests

Some authors are members of the editorial board of Hydrology and Earth System Sciences.

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
