# Peer review of "Quantifying controls on rapid and delayed runoff response in double-peak hydrographs using ensemble rainfall-runoff analysis (ERRA)"

_EGUsphere, 2025_

## Author Response (AR1)

**Response to Reviewers**

**Dear Editor and reviewers:**

Thank you very much for providing constructive comments and helping us to improve our manuscript. We have incorporated most of the suggestions made by the reviewers. Detailed revisions can be found in the tracked changes file.

Due to the revision and addition of new content, the page, line, figure, and equation numbers in the revised manuscript have changed. For clarity, we keep the original numbering used by the reviewers in their comments and indicate the corresponding updated numbers in the revised manuscript (tracked changes file). In our responses, we use the updated numbering.

**Our major revision includes:**

- Newly added equations (2) and (3), presenting more explicit definitions of response distributions GRRDP and RRDGR.
- Newly added Eq. (8) and Figure 3 for NRF, clarifying the calculation of NRF and better distinguish the nonlinear runoff response to precipitation shown in Figure 9 and the nonlinear groundwater recharge response to precipitation shown in Figure 10.
- Refined discussion of the hypothesis.
- More explicit explanations of equipment, calculations, and technical details.

Marks used for different text in this response are:

In shadowed normal text: the reviewer's comments

In black italic text: *original text in the original manuscript*In blue normal text: revised text in the revised manuscript
In black normal text: our response to reviewer's comments

Below we provide a point-by-point response to the individual questions/suggestions.

**Response to Reviewer #1:**

The presented study applies the Ensemble Rainfall Runoff Analysis (ERRA) to explore the mechanisms of double-peak hydrograph emergence in the Weierbach catchment, Luxembourg. The authors test the hypothesis that the double-peak hydrograph is generated by two pathways – a near-surface runoff in direct response to precipitation and a pathway through vadose zone influenced by groundwater recharge. They further explore how double-peak runoff generation varies with precipitation intensities and antecedent catchment wetness. The controls of the first and second hydrograph peak are identified with their specific thresholds. The manuscript thus presents novel insights into the mechanistic functioning of runoff generation inferred with ERRA from multiple observational data in the Weierbach catchment. This is a very interesting study which rigorously works out the controls on the double peak hydrograph in the study catchment and clearly presents the results and in-depth discussion.

**Response: Thank you!**

I have a few remarks that may further enhance the presentation and potentially increase the generalization of the case study results.

Comment 1: L65 - 69 (now L67-72): The cited studies with specific numbers of catchment storage, precipitation volume and intensities are site-specific and very much depend on the catchment characteristics. It would be good to mention this in this paragraph.

**Response:** Thank you for the suggestion. We have specified that in this revised paragraph: The occurrence of double-peak hydrographs has been linked to thresholds of precipitation, antecedent wetness, and catchment storage that are site-specific and depend on catchment characteristics. (Lines 65-67)

Comment 2: With regards to NRF(P) /NRF (GR) the authors define it as a response function of discharge to amount of precipitation/Groundwater recharge in each increment, and this function in its turn depends on precipitation intensity/GR rate at time t. Should this function also depend on the antecedent moisture state as it controls the responsiveness of discharge to P/GR? In case only overland flow as response to P is considered, this is not of large importance, but for GR it should matter, shouldn't it? Or is average soil moisture state implicitly considered in NRF analogously to the rational formula? The runoff coefficient in the rational formula is however an integral over an event duration of a longer time period, but NRF is a function resolved in time depending on precipitation intensity. I somehow miss soil moisture in this resolved representation. Please, explain.

**Response:** Thank you for the question. We assume that the reviewer refers to Figure 9 as "NRF(P)" (original Figure 8) and Figure 10 as "NRF (GR)" (original Figure 9).

Figure 9 shows the nonlinear runoff response to precipitation, where NRF is the product between precipitation-intensity-dependent runoff response function (RRD) and the precipitation rate (as defined in Eq. 6). It quantifies how streamflow responds to one time step of precipitation at a given intensity. Antecedent wetness is treated in ERRA as a category variable, in which the NRF is estimated for two or more different ranges of soil moisture, groundwater levels, antecedent discharge, or antecedent precipitation. The contrast between wet and dry antecedent moisture conditions, as inferred from water table depth, is illustrated by comparing the "wet" and "dry" curves in the figure. In Figure 11, we show that one obtains similar results, whether one infers antecedent wetness from antecedent soil moisture or from antecedent water table depth.

The "NRF(GR)" in Figure 10 is not defined as "a response function of discharge to amount of Groundwater recharge in each increment". Figure 10 shows the **nonlinear groundwater recharge response to precipitation**, where NRF is the product between precipitation-intensity-dependent groundwater recharge response and the precipitation rate. It should be interpreted as the rate of groundwater recharge expected to result at a given time lag from precipitation falling at a specific rate. This coupling between precipitation and groundwater recharge will be sensitive to antecedent wetness, as shown by the two curves in Figure 10, but they do not reflect the influence of soil moisture status on streamflow per se. Again, Figure 11 shows that antecedent soil moisture and antecedent water table depth are nearly equivalent as indicators of antecedent wetness.

We have added Eq. (8) to clarify the difference between NRFs shown in Figure 9 and Figure 10, and specified Eq. (7) and Eq. (8) accordingly in the discussion of Figures 9 and 10 in Line 529 and Line 611, respectively.

Similarly, in a nonlinear system linking precipitation and groundwater recharge, in which the GRRDP at each lag is a function of the precipitation intensity, the NRF is expressed as

$$GR_j = \sum_{k=0}^m \text{NRF}_k(P_{j-k}) \ \Delta t \tag{8}$$

where  $GR_j$  is groundwater recharge at time step j,  $P_{j-k}$  is precipitation occurring k time steps earlier, NRFk is the nonlinear response of groundwater recharge to precipitation that falls at a rate  $P_{j-k}$  and lasts for a time step of  $\Delta t$ , and m is the maximum lag being considered. (Lines 221-226)

**Comment 3:** The results presented in section 3 rigorously demonstrate that the formulated hypothesis that the double-peak hydrograph is generated by the combination of near-surface and groundwater-mediated pathway cannot be rejected. An alternative hypothesis that only e.g., groundwater-mediated pathway would be responsible must be rejected. I think the paper would benefit if the authors more clearly articulate the results in Karl Popper's sense that the formulated hypothesis could not be rejected.

**Response:** Thank you for the suggestion. Two alternative hypotheses are actually rejected in Section 3.3.2: the disconnect between the two curves in Figure 7b (original Figure 6b) rejects the hypothesis that only near-surface pathway can explain the total rainfall-runoff response (because it doesn't explain the second peak), and the disconnect between the two curves in Figure 7c (original Figure 6c) rejects the hypothesis that only groundwater-mediated pathway would be responsible (because it doesn't explain the first peak).

We have made this clear in our discussion of Figure 7 in the revised manuscript:

Figures 7b and 7c also reject two alternative hypotheses. The disconnect between the two curves in Figure 7b rejects the hypothesis that the near-surface pathway alone can explain the total rainfall-runoff response (because it doesn't explain the second peak). Similarly, the disconnect between the two curves in Figure 7c rejects the hypothesis that the groundwater-mediated pathway alone can explain for the total rainfall-runoff response (because it doesn't explain the first peak). (Lines 446-450)

**Comment 4:** L461-465 (now L515-520): Do I understand correctly that in order to analyze the effect of catchment wetness and precipitation intensity on hydrograph generation you need to separate different periods when e.g., WTD falls into three designated classes, and then you estimate NRFs for these periods? Please, explain your methodological steps for clarity.

**Response:** Yes for WTD. For nonstationary analysis (i.e., runoff may respond differently depending on catchment wetness status), we need to separate the precipitation time series into different categories according to the antecedent WTD when rain falls.

In Section 4.1, we estimate RRD for each antecedent wetness category. Comparing RRDs between different categories only examines nonstationary runoff response.

In Section 4.2, we estimate NRFs for each antecedent wetness category. Comparing NRFs within each category only looks at nonlinear runoff response, and comparing NRFs between different categories jointly looks at both nonlinear and nonstationary runoff response.

Thank you for the point. We have clarified this in revised Section 4.1 and Section 4.2:

Lines 484-486: Using the method outlined in Sect. 2.3.4, we compare RRDs between different antecedent wetness categories to quantify how antecedent wetness influences the runoff response of the whole-catchment rainfall-runoff system (i.e., nonstationary runoff response).

Lines 515-520: Here we jointly analyze the influence of precipitation intensity and antecedent wetness on runoff response using the method outlined in Sect. 2.3.4 and Sect. 2.3.5. To quantify how runoff responds to different precipitation intensities under wet vs. dry ambient conditions (i.e., both nonlinear and nonstationary runoff response), we estimate NRFs for both "dry conditions" (the driest 70% of antecedent water table depths, antWTD >1.66 m, which exhibited a single-peak runoff response in Sect. 4.1), and "wet conditions" (the wettest 30% of antecedent WTD values, antWTD ≤1.66 m, which exhibited a double-peak runoff response in Sect. 4.1).

Lines 530-532: The comparison of NRF curves within each antecedent wetness category reflects only nonlinear runoff response, whereas the comparison of NRFs between wet and dry categories jointly reflects both nonlinear and nonstationary runoff response.

**Comment 5:** Is this also done analogously for the precipitation intensity? Precipitation intensity is however, much more volatile than catchment average WTD. How does this affect the results if one maybe need to pick just one or two hours of intensive precipitation out of the entire event. Does ERRA then clearly separates NRFs for one-two hours of intensive precipitation framed by a few hours of less intensive precipitation belonging to another class (and resulting in a different NRF) prior and after a major downpour?

**Response:** No for precipitation intensity. For nonlinear runoff response analysis (i.e., runoff may respond more-than-proportionally to changes in precipitation intensity), NRF equals the precipitation-intensity-dependent RRD times the precipitation rate (Eq. 6), which is "approximated in ERRA by continuous piecewise-linear broken-stick functions of precipitation intensity" (Lines 226-227).

Precipitation is divided into segments between specified precipitation intensity values ("knots"), and the  $NRF_k$  in Eq. (7) (the nonlinear response of streamflow to precipitation that falls at a rate  $P_{j-k}$  and lasts for a time step of  $\Delta t$ ) is the sum of these segments, each multiplied by the slopes of the corresponding broken-stick segments (Kirchner 2024a). That is, NRF is an estimate of the ensemble average of the responses to many rainfall events instead of only over an individual hour or two at a given precipitation intensity.

Thank you for comments 4 and 5. For clearer clarification of the calculation of the Nonlinear Response Function (NRF), we have added a new Figure (Fig. 3) in Section 2.3.5:

Figure 3. Schematic illustration of how a runoff response distribution (RRD) characterizes linear runoff response at a given lag k (a), and how a nonlinear response function (NRF) characterizes nonlinear runoff response (b). Grey points indicate how one time step of precipitation at a rate P alters discharge Q at a lag k. (ERRA statistically corrects these points for the overlapping effects of other precipitation inputs at other time lags, making them analogous to leverages in multiple regression.) If the runoff response is approximately linear, it can be approximated by the dashed line in (a), the slope of which is the RRD for that lag. If the runoff response is nonlinear, it can be approximated by a piecewise-linear relationship such as the dashed line in (b), connecting a series of knot points (open circles) at precipitation rates  $\kappa_0$ - $\kappa_4$ . Such relationships are functions of P and thus cannot be characterized by single values, like RRDs can. The NRF and the RRD have different dimensions because NRF estimates the effect of P on Q (the abscissa of (b), whereas the RRD estimates the slope of the relationship between P and P's effect on Q.

**Comment 6:** The analysis of nonstationarity and nonlinearity in section 4 is very interesting, but sensitivities expressed in absolute numbers remain pertinent to the study catchment. I am wondering if one could derive some dimensionless or relative measures that can be generalized when investigations from a large set of catchments would be available. Would it be helpful for example to look not at the catchment average WTD but in relation to the mean annual precipitation?

**Response:** Thank you. We are indeed conducting a large-sample study of nonlinearity and nonstationarity, but that would be a completely different analysis. An example of large set of catchments study looking at WTD's influence on runoff response can be found here for your reference:

Eslami, Z., Seybold, H., and Kirchner, J. W.: Climatic, topographic, and groundwater controls on runoff response to precipitation: evidence from a large-sample data set, EGUsphere [preprint], https://doi.org/10.5194/egusphere-2025-35, 2025.

Looking at the relationship to mean annual precipitation may work in inter-catchment comparisons, but not here in the nonstationary analysis of our manuscript because the single value of mean annual precipitation in a catchment cannot be used to split the precipitation time series into different categories in the nonstationary response analysis or to divide different precipitation intensity ranges in the nonlinear response analysis.

**Comment 7:** I understand that results presented in section 4 are catchment specific and much of the discussion links to previous studies and mechanisms of runoff generation in this very catchment, but still maybe some ideas for generalizations may enrich the discussion part from L539ff (now L596ff).

**Response:** Thank you for the point. We presented a site-specific discussion because we are reluctant to generalize beyond the data that we actually have, and have actually analyzed.

We now added this idea after the discussion part from L603: These discussed behaviors are specific to the runoff generation mechanism in the Weierbach catchment, based on the data we actually have, and have actually analyzed. Although the inferred runoff mechanisms are not amenable to generalization yet, the methods and hypotheses may provide useful insights for explorations in other catchments, and in inter-catchment comparison studies. (Lines 603-607)

Comment 8: Finally, the potential artifact with near-zero spike mentioned in section 3.1 seems to remain unclear. It says in L234 (now L276) it will be explored in the next section, but I somehow missed a detailed analysis. Do you mean that the exploration is given in L274-277 (now L319-322) which basically concludes that you do not have enough data to pinpoint the origin of this spike? Or did I miss further elaboration on this issue in the manuscript? If there is not enough data to further explore this artifact, I suggest not to raise expectation in L234 (now L276).

**Response:** The exploration of the potential artifact with near-zero spike was indeed given in L319-322 (original L274-277). This artifact first appears in Figure 4c, where we inferred that it could potentially be caused by the distortion of the strong short-lag relationship between groundwater recharge and precipitation (observed in Figure 4b) or the rapid runoff effects of groundwater recharge in the near-stream zone. We then explored how the spike can be reduced by jointly analyzing the mixing effects of correlated precipitation and groundwater recharge on streamflow (Figure 5). To resolve the source of the remaining spike would require data that unfortunately don't exist.

Thank you for the suggestion. We have revised the Line 276 to "In the following section, we further explore how this potential artifact can be reduced by jointly analyzing the effects of correlated precipitation and groundwater recharge on streamflow." to avoid

raising expectation for resolving the artifact.

**Response to Reviewer #2:**

Comment 1: The manuscript is focusing on the streamflow analysis with Kirchner's ERRA- method. The catchment reacts depending on the antecedent conditions with or without a double peaked runoff response. The authors describe the method and the applicability of it. The influence of precipitation intensity is shown and the effect of different antecedent wetness measures are shown. The manuscript delivers a simple analysis method, with which these complex runoff responses could be estimated and is therefore an important contribution for the scientific community for runoff response in small to medium sized hydrological catchments for quantitative but as well for qualitative perspectives.

**Response:** Thank you for your support.

**Comment 2:** Because in the second peak could be substances transported which were remobilised like plant protection products, fertilisers and their metabolites, etc. depending of the pH values and the solutability.

**Response:** We appreciate this point. We have added this perspective to the introduction and interpreted it more generally:

Furthermore, understanding the processes underlying both peaks may be important because different flowpaths may transport different potential contaminants. (Lines 78-79)

**Comment 3:** There are several parts which could be moved to an appendix. And the important topic of measures which could be used as antecedent conditions measures is missing in the introduction. The structure is confusing. The rainfall intensity is quite dominant. The antecedent condition measures have only a small part of the manuscript but are quite important. In chapter 5 it is not clear how they were considered no equation is presented for soil moisture, and antecedent runoff.

**Response:** We appreciate the reviewer's enthusiasm concerning measures of antecedent conditions, but that is not the focus of our paper. ERRA is a method for data analysis, not a simulation model, so it is not straightforward to declare one or another measure of antecedent conditions as the "best", even at this one site, and any such result would not be transferable to other sites. In Section 5.2, we show that all antecedent water table depth, antecedent soil moisture, and antecedent discharge all yield broadly similar results at this site. We also point out that these different proxies for antecedent conditions have different response times, so that the choice among them (in cases where one is lucky enough to have any such choice at all) will depend on the question one is

trying to answer.

We did not present an equation for soil moisture or antecedent runoff because these are just lagged values from the soil moisture and discharge time series. This is indicated in Line 486: "...we use the antecedent catchment-averaged water table depth (antWTD) measured 6 hours before precipitation falls..." and Line 649: "...except when antQ is measured 24 hours before precipitation falls...".

Thank you for the point. We have pointed this out in the revised description more explicitly: We tested each of these antecedent wetness proxies crossed with four categories of antecedent time lag (antWTD, antVWC, and antQ measured 1, 6, 12, and 24 hours before precipitation falls)" (Lines 637-639)

**Comment 4:** The mathematical description of the approach is separated into two parts and should be presented in one block. Figure 3 and 4 (now Figures 4 and 5) are results and should be moved to 3.3.2 or to the appendix.

**Response:** It appears that our use of "Result" as the heading for Section 3.3.2 has created the impression that all of the results are there, or should be there, whereas this is only the result of the hypothesis test posed in Section 3.3.1. In fact, Sections 3 and 4 (and Figures 4-9) are all results, with Section 5 and Figures 10-11 extending the results (which is the point of a discussion section).

Thank you for pointing this out. We have changed the heading for Section 3.3.1 from "*Hypothesis*" to "Two-pathway hypothesis", and changed the heading for Section 3.3.2 from "*Result*" to "Result for two-pathway hypothesis".

What the reviewer refers to as "the mathematical description of the approach" is actually two different things, which would be very confusing if they were presented "in one block". The first, which is presented in Section 2.3, is an overview of Ensemble Rainfall-Runoff Analysis, which is essential background for everything that follows. Readers then need to see the results from this analysis (Figures 4 and 5) before they are in a position to understand the motivation behind the hypothesized two pathways underlying the double-peak hydrograph, and the convolution model for formally testing that hypothesis (Section 3.3 and Figure 6). It would not be an effective communication strategy to present this convolution model before readers had any idea what it was needed for.

**Comment 5:** I would suggest a complete reorganisation of the manuscript and more clear formulation of the hypotheses.

**Response:** The organizational structure of the manuscript grew out of careful consideration of what readers need to know at each point in the paper (as illustrated, for example, by our response to Comment 4 above). Many "quick fixes" that seem superficially attractive would be counterproductive in practice (like, as discussed above, presenting all the math "in one block").

Regarding the hypothesis, we have stated in our revised version that Figure 7 also tests two alternative hypotheses to have a more explicit formation of the hypotheses:

Figures 7b and 7c also reject two alternative hypotheses. The disconnect between the two curves in Figure 7b rejects the hypothesis that the near-surface pathway alone can explain the total rainfall-runoff response (because it doesn't explain the second peak). Similarly, the disconnect between the two curves in Figure 7c rejects the hypothesis that the groundwater-mediated pathway alone can explain for the total rainfall-runoff response (because it doesn't explain the first peak). (Lines 446-450)

**Comment 6:**

**Abstract:**

Add the different used antecedent conditions measures and which gave the best results.

**Response:** Thank you for the suggestion. As indicated above, the purpose of the paper is not to compare different measures of antecedent wetness, and there is no clear standard to assess one or another as being "best". We have revised the description in the abstract to make the measures of antecedent wetness in our main results more explicit:

Under wet conditions (here defined as antecedent water table depth  $\leq 1.66$  m), the first peak increases nonlinearly (particularly at precipitation intensities above 2 mm h-1) and the second peak becomes higher, narrower, and earlier with increasing precipitation intensity. Under dry conditions (here defined as antecedent water table depth > 1.66 m), the first peak increases nonlinearly with precipitation intensity (particularly above 4 mm h-1), and groundwater recharge also responds to precipitation, but no clear second peak occurs regardless of precipitation intensity. (Lines 21-25)

**Comment 7:**

**Introduction:**

Add which antecedent wetness conditions could be used which is a crucial measure to detect the double peak phenomena (antecedent precipitation index, antecedent soil moisture index, antecedent groundwater, and pre- event runoff)

**Response:** Thank you for the point. Antecedent wetness proxies are not, in fact "crucial... to detect the double peak phenomena". For example, Figures 4-7 clearly

illustrate the double-peak phenomenon without any information about antecedent wetness.

**Comment 8:** Comparison of different proxies for catchment antecedent wetness conditions

**Response:** We are not sure what is meant here. Figure 11 compares the results obtained with different antecedent wetness proxies.

**Comment 9:** I was expecting that the authors present a threshold value at which double peaks occur.

**Response:** Thank you for the point. We are reluctant to specify a particular threshold for the emergence of double peaks, for several reasons. The occurrence of double peaks depends on both antecedent wetness and precipitation intensity. The general tendencies in these relationships are clear; as Figure 11 shows, the second peak is obvious when water table depth is between 1.66 and 1.30 m, but negligible when water table depth is below 1.66 m. And it is more obvious at higher precipitation intensities. But it is difficult to define a specific threshold at which the second peak emerges (vs. at which it is present but very small). And even if we arbitrarily define a size threshold for the second peak (at which it would be declared to be "present" rather than "absent"), any corresponding threshold of antecedent wetness would be specific to Weierbach and not amenable to generalization.

**Comment 10:** The authors should explain why they have selected antecedent groundwater table and what speaks against soil moisture and pre- runoff conditions.

**Response:** We used antecedent water table depth (a) because of the evident role of groundwater in generating the second peak, (b) because the available soil moisture measurements only reflect the wetness state of the upper 60 cm of the subsurface, and (c) because the interpretation of antecedent streamflow necessarily depends on whether that streamflow results from the first peak (which is driven primarily by precipitation intensity) or the second peak (which is driven primarily by catchment wetness, and more specifically groundwater).

Thank you for pointing this out. We have added an explanation about this in Section 5.2:

Catchment antecedent wetness conditions can be described by a range of proxy measurements. For example, available proxies at Weierbach include antecedent water table depth (antWTD), antecedent volumetric water content (antVWC), and antecedent

streamflow (antQ). Soil moisture measurements here only reflect the wetness state of the upper 60 cm of the subsurface (Fig. 2). The interpretation of antecedent streamflow necessarily depends on whether that streamflow results from the first peak (which is driven primarily by precipitation intensity) or the second peak (which is driven primarily by catchment wetness, and more specifically groundwater). Therefore, considering the evident role of groundwater in generating the second peak, we used antWTD as a proxy for antecedent wetness in our assessment of nonstationary runoff response to precipitation in Sect. 4. (Lines 614-620)

**Comment 11:** Explain why the lowest soil moisture probe was selected and not a mean value of the probes.

**Response:** The assumption made here is incorrect. Our analysis used all the available probes. Line 654 says explicitly that antecedent VWC reflects soils  $\leq$  60 cm (i.e., the entire depth range of all the soil moisture probes).

Thank you for the point. We have added more explicit explanation in Section 5.2:

In this section, we compare antWTD with antVWC and antQ as alternative proxies for catchment-scale antecedent wetness in runoff response analyses. WTD is averaged among three available wells, reflecting the catchment wetness status at depths of ~0.6–3 m. VWC is averaged among all available probes at all depths, reflecting the catchment wetness status of the unsaturated zone in the upper 60 cm. Streamflow itself reflects the integrated catchment wetness status. (Lines 634-637)

**Comment 12:** If soil moisture would be equivalent wouldn't it be the better proxy because installing probes is easier to install and less cost intensive?

Response: Thank you. We are not prepared to make such categorical value judgments. Soil moisture probes will be a better proxy if the most mechanistically relevant antecedent wetness is soil moisture rather than groundwater storage. Whether soil moisture probes are easier to install and less cost intensive will depend on how many of them are needed to adequately capture the spatial heterogeneity in soil moisture (which will not be known in advance). Conversely, the effort and expense of monitoring groundwater will depend on how heterogeneous the substrate is and how difficult it is to drill, and on how large the effective footprint of each well is (which will also not be known in advance).

In practice, researchers are likely to use whatever antecedent wetness proxies are available, which is why we undertook to compare them in Section 5.2, taking advantage of the fact that all three of them are available at Weierbach.

**Comment 13:**

Specific comments:

Page 4 Figure 1: which GW gauge was used for the analysis?

**Response:** The manuscript already explains that we used the average of all three wells. Lines 132-134 say that "we used the three wells with the most complete records covering the upper plateau, the middle of the hillslopes, and low hillslope positions in the catchment (GW2, GW3, and GW5; Fig. 1)". Line 144 says that we averaged the changes in WTD across all three wells to infer groundwater recharge by "...and then averaging the three wells to obtain the catchment-average GR". And line 486 says that "...we used the catchment-averaged water table depth..." as a proxy for antecedent wetness.

Thank you for the comment, we have added a note in Section 2.2 (Lines 151-152) to make these points more explicit: Catchment-averaged variables (average GR and average WTD of three wells, average VWC of all probes at all depths) are used in the analyses presented here.

Comment 14: Page 5 Line 127 (now Line 135) which type of device was used

**Response:** Soil moisture is recorded on CR800 loggers using CS650 water content reflectometers (Campbell Scientific, Logan, Utah, USA) (Hissler et al., 2021).

Detailed descriptions of equipment and data collection can be found in Hissler (2021) (Line 118). We have noted this in Lines 134-136: Volumetric soil water content was measured at 10-cm, 20-cm, 40-cm, and 60-cm depth at five sites (Fig. 1) every 30 min using CS650 water content reflectometers (Campbell Scientific, Logan, Utah, USA).

**Comment 15:** Page 6 Line 145 (now Line 154) is the presented VWC the mean value for all soil moisture monitoring points? Or is it a specific point in figure 1?

**Response:** At each depth it is the average of all available probes.

Thank you for the question. We have revised the caption of Figure 2 to: Figure 2. Overview of measured time series of precipitation (P), volumetric water content (VWC) for 4 depths (average of all available probes at each depth)...

Comment 16: Page 9 Line 207 (now Line 250): RRDP is defined in eq. 2 (now Eq. 4)

**Response:** Not quite. Equation 4 defines the partial RRD, partialRRDP, discussed in Section 3.2.

RRDP and GRRDP are two different versions of the RRD defined in Equation 1. If the input is precipitation and the output is discharge, the RRD is denoted RRDP (the distribution of runoff response to precipitation). Alternatively, if the input is precipitation and the output is groundwater recharge, the RRD is denoted GRRDP (the distribution of groundwater recharge's response to precipitation).

Although this is already explained at the beginning of Sections 3.1 and 3.2, we now see how readers could miss it, so we have added explicit definitions of GRRDP and RRDGR in Section 2 (along with new added equations 2 and 3 for them):

**2.3.2 Groundwater recharge response distribution driven by precipitation (GRRDP)**

Analogously to Sect. 2.3.1, an unsaturated zone system linking a single precipitation input (P) and a single (or spatially averaged) groundwater recharge output (GR) could potentially be approximated as a convolution with discrete time steps of length  $\Delta t$ :

$$GR_j = \sum_{k=0}^{m} GRRD_{P,k} P_{j-k} \Delta t$$
 (2)

where  $GR_j$  is groundwater recharge at time step j,  $P_{j-k}$  is precipitation occurring k time steps earlier,  $GRRD_{P,k}$  is the impulse response of groundwater recharge to precipitation at lag k, and m is the maximum lag being considered.

**2.3.3 Runoff response distribution driven by groundwater recharge (RRDGR)**

Analogously to Sect. 2.3.1, a saturated zone system linking a single (or spatially averaged) groundwater recharge input (GR) and a single streamflow output (Q) could potentially be approximated as a convolution with discrete time steps of length  $\Delta t$ :

$$Q_j = \sum_{k=0}^{m} RRD_{GR,k} GR_{j-k} \Delta t$$
 (3)

where  $Q_j$  is streamflow at time step j,  $GR_{j-k}$  is groundwater recharge occurring k time steps earlier,  $RRD_{GR,k}$  is the impulse response of streamflow to groundwater recharge at lag k, and m is the maximum lag being considered.

**Comment 17:** Page 10 Line 219 (now Line 261) equation is missing for GRRDP - it is presented at page 14 Line: 341 (now Line 387)

Response: Thank you. We have added explicit definitions of RRDP, GRRDP, and

RRDGR in Section 2, as shown in reply to Comment 16 above.

**Comment 18:** Page 14 Line 350- 352 (now Line 396-398): equation 10 or 11 (now Eq. 13 or Eq. 14) could be removed

**Response:** We disagree. Equation 13 follows from Equations 11 and 12, and without explicitly stating that convolution is associative one cannot get from Equation 13 to Equation 14. Equation 14, in turn, is necessary as a component of Equation 16.

**Comment 19:** Page 25 Lines: 566-571 (now Line 637-643): Is it important to know the percentiles?

**Response:** We think so. Otherwise it is difficult for readers to understand the ranges of catchment conditions corresponding to the plots in Figure 11.

**Comment 20:** Page 27 Line 622 (now Line 691) This is exactly the same conclusion like in the Schaefertal catchment (Graeff et al. 2009) where the double peak events could not be any more observed after mining activities below the catchment started and groundwater was completely disturbed

**Response:** We disagree that this is "exactly the same conclusion", since draining a catchment from below by mining is different from depleting groundwater by evapotranspiration and stream discharge. It is also not correct to say that double-peak events "could not be any more observed after mining activities" at Schaefertal, because Graeff et al. (p. 705) note that double-peak events were still observed "in response to very strong precipitation events".

However, we appreciate the pointer to this interesting study, which we were previously unaware of. We have included this study in the introduction:

In an interesting historical example, double-peak hydrographs were commonly observed in the Schaefertal (Germany) in the 1970's, but became rare, only occurring in response to intense precipitation, after mining activities commenced below the catchment, leading to groundwater depletion by mine drainage (Graeff et al., 2009). (Lines 72-74)

**Response to Editor (Roger Moussa):**

**Comment 1:** The paper applies the Ensemble Rainfall Runoff Analysis (ERRA) to explore the mechanisms of double-peak hydrograph. Applications were conducted in the Weierbach catchment, Luxembourg. This is a very interesting study presenting novel insights into the mechanistic functioning of runoff generation. The paper is clear and well structured.

Response: Thank you.

Comment 2: Two Reviewers evaluated the paper and gave detailed comments. I agree with the evaluation of both Reviewers who stated that the topic of international interest, but both Reviewers suggest recommendations to enhance the presentation and potentially increase the generalization of the results, especially Reviewer #2 who suggests a complete reorganisation of the manuscript and more clear formulation of the hypotheses. In their responses, the authors suggest improvements to the manuscript.

For these reasons, I recommend major revision and invite the Authors to modify the paper responding point by point to all points raised by both Reviewers.

**Response:** Thank you. We have incorporated the suggestions made by the reviewers in our revised manuscript and responded to all comments point by point above. We have newly added equations (2), (3), and (8), and Figure 3 to better present the calculation and differences between different response distributions and nonlinear response functions (NRFs). We have refined our discussion of the hypothesis and included more details of equipment, calculations, and technical details to more explicitly clarify and present our results.